# Spatial Variability of Beach Impact from Post-Tropical Cyclone Katia (2011) on Northern Ireland's North Coast

**Giorgio Anfuso [1],\*, Carlos Loureiro [2,3], Mohammed Taaouati [4], Thomas Smyth [5] and Derek Jackson [6]**

1   Department of Earth Sciences, Faculty of Marine and Environmental Sciences, University of Cádiz, Polígono del Río San Pedro s/n, 11510 Puerto Real, Spain
2   Biological and Environmental Sciences, Faculty of Natural Sciences, University of Stirling, Stirling, Scotland FK9 4LA, UK; carlos.loureiro@stir.ac.uk
3   Geological Sciences, School of Agricultural, Earth and Environmental Sciences, University of KwaZulu-Natal, Durban 4001, South Africa
4   Department of Exact Sciences, National School of Architecture, Tetouan 93000, Morocco; mtaaouati@gmail.com
5   Department of Biological and Geographical Sciences, School of Applied Sciences, University of Huddersfield, Huddersfield HD1 3DH, UK; T.AG.Smyth@hud.ac.uk
6   School of Geography & Environmental Sciences, Ulster University, Coleraine, Co. Londonderry, Northern Ireland BT52 1SA, UK; d.jackson@ulster.ac.uk
\*   Correspondence: giorgio.anfuso@uca.es; Tel.: +34-956016167

**Abstract:** In northern Europe, beach erosion, coastal flooding and associated damages to engineering structures are linked to mid-latitude storms that form through cyclogenesis and post-tropical cyclones, when a tropical cyclone moves north from its tropical origin. The present work analyses the hydrodynamic forcing and morphological changes observed at three beaches in the north coast of Northern Ireland (Magilligan, Portrush West's southern and northern sectors, and Whiterocks), prior to, during, and immediately after post-tropical cyclone Katia. Katia was the second major hurricane of the active 2011 Atlantic hurricane season and impacted the British Isles on the 12–13 September 2011. During the Katia event, offshore wave buoys recorded values in excess of 5 m at the peak of the storm on the 13 September, but nearshore significant wave height ranged from 1 to 3 m, reflecting relevant wave energy dissipation across an extensive and shallow continental shelf. This was especially so at Magilligan, where widespread refraction and attenuation led to reduced shore-normal energy fluxes and very minor morphological changes. Morphological changes were restricted to upper beach erosion and flattening of the foreshore. Longshore transport was evident at Portrush West, with the northern sector experiencing erosion while the southern sector accreted, inducing a short-term rotational response in this embayment. In Whiterocks, berm erosion contributed to a general beach flattening and this resulted in an overall accretion due to sediment influx from the updrift western areas. Taking into account that the post-tropical cyclone Katia produced £100 m ($157 million, 2011 USD) in damage in the United Kingdom alone, the results of the present study represent a contribution to the general database of post-tropical storm response on Northern European coastlines, informing coastal response prediction and damage mitigation.

**Keywords:** wave energy; Hurricane Katia; longshore transport; dissipative

## 1. Preamble: Katia Cyclone Description

Hurricane Katia's formation was instigated by a wide low-pressure system on the 28th of August 2011, offshore of the western coast of Africa (Figure 1). The low pressure system moved westward and, on 29 August, acquired sufficient convective intensity to be designated as a tropical depression when it was located about 695 km southwest of the south westernmost Cape Verde Islands (https://www.nhc.noaa.gov, accessed on 18 April 2020). The depression moved to the west-northwest for the next 24 h and gradually strengthened, becoming a tropical storm on 30 August about 787 km southwest of the Cape Verde Islands. The cyclone maintained a west-northwest trajectory at around 27.8 km h$^{-1}$ for the next two days and steadily strengthened to reach hurricane intensity on the Saffir-Simpson Hurricane Wind Scale by 1 September when it was located about 2176 km east of the Leeward Islands. After achieving hurricane status, Katia turned northwest and continued to strengthen and reached hurricane category 4 status on 5 September with wind peak intensity of 220 km h$^{-1}$ and a central low pressure of 942 mb when the hurricane was located about 870 km south of Bermuda. Such extreme conditions lasted one day only and the hurricane then slowed down and gradually turned north-east on 9 September (Figure 1). After this, the wind field expanded and weakened. When Katia was located about 650 km northwest of Bermuda, it turned toward the east-northeast and increased in speed, to approximately 92 km h$^{-1}$, arriving over the cold sea-surface temperatures (22 °C) of the North Atlantic Ocean. The combination of cold water and strong vertical wind shear favoured the quick transition from a hurricane status into a powerful post-tropical low-pressure system by 1200 UTC 10 September when it was located about 463 km south-southeast of Cape Race, Newfoundland. On 11 September, Katia cyclone, a large and powerful post-tropical storm, turned north-east towards the northern British Isles with an average velocity of 85 km h$^{-1}$. The post-tropical cyclone reached the northern coast of Scotland on 12 September and produced sustained gale-force winds across most of the British Isles and hurricane-force wind gusts in Scotland, Northern Ireland, and northern England with average wind speed from 101 to 188 km h$^{-1}$ with peak values of 212 km h$^{-1}$ recorded in North Wales. On the 13 September the cyclone continued north-eastward and dissipated over the North Sea. In Europe, the post-tropical cyclone Katia impacted numerous locations, downing trees, bringing down power lines, and leaving thousands without electricity. In the United Kingdom the storm was responsible for two deaths and caused approximately £100 m ($157 million 2011 USD) in damage [1].

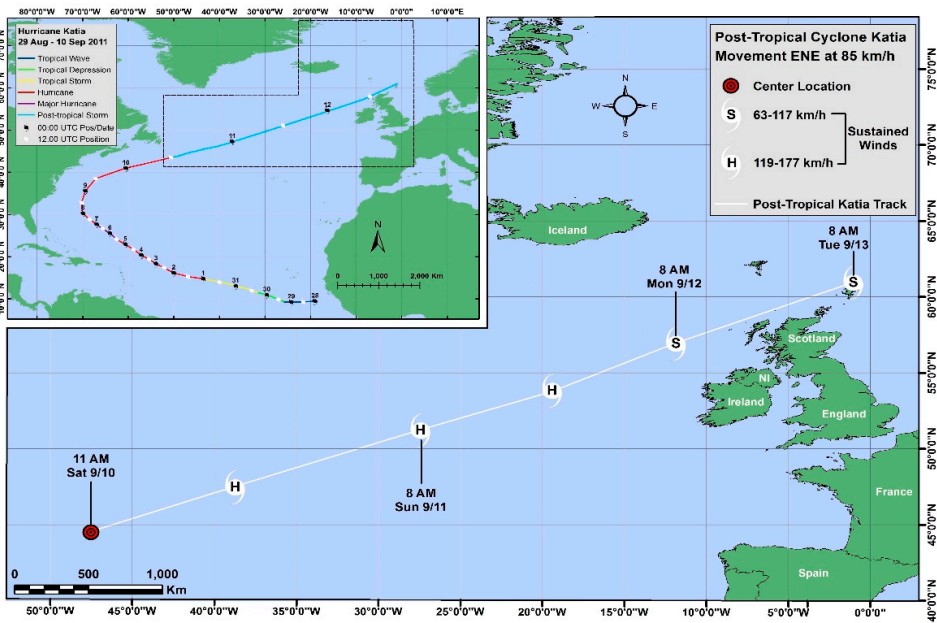

**Figure 1.** Track of Katia cyclone obtained from NOAA [1].

## 2. Introduction

Coastal development continues to increase and some 50% of the world's coastline is currently under pressure from excessive development [2,3], mainly in the form of tourism, one of the world's largest industries [4,5]. In Europe, the rapid expansion of urban artificial surfaces in coastal zones during the 1990–2000 period [6], has occurred in the Mediterranean and South Atlantic areas, namely Portugal (34% increase) and Spain (18%), followed by France, Italy and Greece. Ireland, a more peripheral holiday beach destination, has also had significant development of urban artificial surfaces in the coastal zones. In Northern Ireland, in 2018, 2.8 million visitors and over 2 million Northern Ireland residents took an overnight trip to the region, spending an unprecedented £968 million, £42 million more compared with 2017 [7].

Activities related with tourism can be significantly affected by the impacts of storms and hurricanes, producing damages to recreational and protective structures, with associated reduction of beach width and general aesthetics [8]. Over the past century, several storms and hurricanes have caused vast economic loses along with large numbers of deaths along the world's coastlines [9]. According to Dolan and Davis [10], the most powerful storms that have struck the Atlantic coast of USA, occurred on the 7–9 March 1962 (the "Ash Wednesday" storm), 7–11 March 1989, and on the 28 October–1 November 1991 (the "All Hallows' Eve").

Morton and Sallenger [11] investigated hurricanes and tropical cyclones that impacted the Gulf of Mexico and the Atlantic coast of USA, evaluating damages and washover penetration linked to hurricanes Carla (1961), Camille (1969), Frederic (1979), Alicia (1983) and Hugo (1989). According to Sallenger [12], the greatest coastal changes are recorded when the beach system completely submerges in an inundation regime: it can take place locally on a barrier island, incising a new inlet, as happened during Hurricane Isabel (2003) in North Carolina, Charley and Ivan in Florida (2004), and Katrina (2005) in Alabama. Additionally, inundation can submerge tens of kilometres of coast as occurred on the Bolivar Peninsula (TX, USA) during hurricane Ike (2008) and in Louisiana during Rita and Katrina events in 2005.

In northern Europe, damages to coastal structures, beach erosion and flooding inundation are often associated with mid-latitude (or extratropical) storms that form through cyclogenesis in the mid-latitude westerly wind belt and, secondly, post-tropical cyclones that form when a tropical cyclone moves north from its tropical origin [13].

In terms of the effects of mid-latitude storms, Bonazzi et al. [14] reconstructed spatial maps of peak gust footprints for 135 of the most important damaging events in 15 European countries in the past four decades, i.e., 1972–2010. The most important storms were 87J, Daria, Vivian, Anatol, Lothar, Martin, Erwin/Gudrun, Kyrill, Emma, Klaus and Xynthia. They observed 64% of events used in their analysis occurred during North Atlantic Oscillation positive phase (NAO+) months and their inter-annual variability, described by the NAO, modulated the main orientation of the storm tracks and the frequency of storm events.

On the Atlantic edge of Europe, Anfuso et al. [15] characterized, using the Storm Power Index [10], the distribution of storms in the Gulf of Cadiz during the 1958–2001 period and, highlighted particularly stormy years, e.g., years characterized by more storms and extended storm durations. Rangel-Buitrago and Anfuso [16] and Anfuso et al. [15] also observed the most powerful stormy years in Cadiz occurred every 5–6 years (e.g., in 1995–1996, 2002–2003, 2009–2010) with a 7–8-year periodicity recorded by Ferreira et al. [17] and Almeida et al. [18] in Faro (Southern Portugal).

The energetic conditions recorded in the Cadiz Gulf area during the 1995–1996 period also corresponded with similar weather conditions observed over the same period in Wales (UK) by Phillips [19] and Phillips and Crisp [20]. Dodet et al. [21] highlighted a highly unusual sequence of extratropical storms over the 2013–2014 winter period along Europe's Northeast Atlantic region, incorporating wave analysis over the period 2002–2017 for the northwest of Ireland, the Bay of Biscay and west of Portugal and beach erosion/recovery in five beaches in SW England, Brittany and the Bay of Biscay (France) that were surveyed on a monthly basis for more than 10 years. That winter recorded

the most energetic conditions along the Atlantic coast of Europe since at least 1948 [22] resulting in most of western Europe' coastlines being severely impacted [23–25] albeit with the exception of beaches in NW Northern Ireland (this field site).

Santos et al. [26] examined waves around UK that exceeded the 1 in 1-year return level analysed from 18 different buoy records for the period from 2002 to mid-2016, de-clustered into 92 distinct storm events. The majority of events were observed between November and March, with large inter-annual differences in the number of events per season associated with the West Europe Pressure Anomaly. The 2013/2014 storm season represented an outlier in terms of the number of wave events, their temporal clustering and return levels.

The strength and position of extratropical cyclones is influenced largely by the pattern of atmospheric circulations over the North Atlantic Basin, which, in turn, are reflected in the signal and strength of the North Atlantic Oscillation Index [27]. However, Atlantic tropical cyclones that move northward from the tropics and undergo extratropical transition may also cause high-impact weather events in Western Europe [28]. Tropical cyclones generated in the Atlantic basin drift westward at tropical latitudes within the easterly Trade winds, and migrate northward affecting the east coast of the US. Usually once every 1–2 years, these cyclones move eastward undergoing extratropical transition and reach western Europe as post-tropical storms often with hurricane force-winds [29]. In Northwest Ireland, the earliest reported high-magnitude event was the 'Night of the Big Wind', which was probably the tail-end of a hurricane reported in January 1839. Cooper and Orford [30] described the occurrence and impacts of post-tropical cyclones on the British Isles using historical and contemporary information. Examining the period between 1922 and 1998, they identified nine major tropical cyclones that traversed the Atlantic and impacted the British Isles. MacClenahan et al. [31] identified Hurricane Debbie (September 1961) as the largest storm that impacted Ireland during the second half of the 20th Century. Recently, Guisado-Pintado and Jackson [32,33] described the effects of the post-tropical Storm Ophelia (2018) and Storm Hector (2019) in Ireland's NW Donegal coast.

The post-tropical cyclone Katia impacted the British Isles during the 12–13 September 2011 causing £100 m ($157 million, 2011 USD) in damage. The present work analyses what, if any, morphological changes occurred during the Katia cyclone in three beaches in the north coast of Northern Ireland, taking into account that storm-induced waves persisted until the 15 September, a couple of days after the cyclone dissipation. The importance of the present study lies in the necessity of understanding and predicting morphological changes associated with the impacts of hurricanes and intense post-tropical storms that, while infrequent, can sometimes have significant impacts on exposed coastal areas of the British Isles and cause relevant economic losses [30]. The behaviour of these coastal systems can be greatly affected in the future due to observed and modelled changes in frequency and intensity of extreme storms, and particularly the poleward migration of the maximum intensity of tropical cyclones as a result of global climate changes [34]. There is also concern that a possible change in hurricane tracks could lead to such destructive events impacting more frequently Southern European coasts, resulting in potentially more dramatic responses [35,36]. The results of the present study contributes to our understanding of beach and coastal response to post-tropical storm events along the coast of Northern Ireland and adds information to the general database of storm response on coastlines of this nature, informing damage mitigation and coastal response prediction.

## 3. Study Area

This paper examines the morphological change in three sandy beach sites, Magilligan Strand, Portrush (West Strand) and Whiterocks (eastern section of Curran Strand) (Figure 1, Table 1). These beaches are located on a high wave energy, microtidal, 20 km section of Northern Ireland's northern coastline [37,38]. Magilligan Strand, the most westerly beach studied, is part of a 10 km long, dissipative beach that stretches from Magilligan Point in the west to Downhill in the east. The area of beach monitored at Magilligan has been accreting since 1980 [39] and is backed by large dunes, approximately 10 m in height, which are densely vegetated with *Ammophila arenaria*.

**Table 1.** Main attributes of selected sites.

| Site Name Location | Sand Grain Size (mm) | Beach Slope (tan β) | Tidal Range (m) | Beach Type | Local Geomorphology |
|---|---|---|---|---|---|
| Magilligan | 0.17 | 0.0375 | 1.6 | Intermediate to dissipative | Extensive dune systems, tidal inlet, sand ridge plain |
| Portrush northern | 0.186 | 0.0320 | 1.5 | Dissipative | Modified dunes, human modification (sea wall) along coastline |
| Whiterocks | 0.197 | 0.0352 | 1.5 | Intermediate | Extensive dune system behind beach |

West Strand (Portrush) is an 850 m long, concave shaped beach bounded by basalt headlands to the northeast and southwest. Two sectors have been investigated within this pocket beach, one at the northern and one at the southern part and respectively noted as 'Portrush southern' and 'Portrush northern' sectors. The beach has undergone significant development beginning in 1825 when a jettied harbour was constructed against the north eastern headland [40]. In the 1960s, a promenade and car park were constructed on the dune complex behind the beach and a recurved seawall constructed on the back beach replaced the natural foredune. This development resulted in significant lowering of the beach surface elevation [41].

Whiterocks beach is located at the easternmost extremity of Curran Strand and is the most easterly study site investigated. Curran Strand is a 3 km long beach constrained by a basalt headland to the west and chalk cliffs to the east. The beach is convex in shape due to the sheltering effect of the Skerries islands located approximately 1.5 km offshore, however wave refraction around the islands produces high energy waves at the eastern extremity of the beach. The section of beach monitored at Whiterocks is backed by chalk boulders and a single steep vegetated foredune ranging from 6–25 m in height behind which a golf course has been constructed.

High-resolution multibeam bathymetric data for this coastline, collected in the framework of the Joint Irish Bathymetric Survey completed in September 2008, demonstrates an irregular and dynamic configuration of the continental shelf and shoreface of the North Coast of Northern Ireland, with tidal banks and sand waves indicating complex flow patterns and active sediment transport pathways (Figure 2). The substratum of the shelf and shoreface of this coastal area is predominantly composed of fine to medium sand sediments, with most exposed bedrock and stony outcrops close to the shore [42]. A wide and relatively flat shoreface extends for over 6 km with depths of less than 15 m offshore Magilligan beach, flanked by the Tuns Bank, a large ebb-delta associated with the Foyle River. The shoreface of Portrush beach is much narrower and steep, with a relatively linear configuration and reaching depths in excess of 18 m approximately 1.2 km seaward of the beach. The offshore shelf and shoreface at Whiterocks presents a complex configuration, influenced by the presence of the Skerries islands and their influence on wave, tidal and sediment transport fluxes. The most exposed section of Whiterocks shoreface is relatively similar to Portrush beach, reaching depths in excess of 20 m approximately 2 km seaward of the beach.

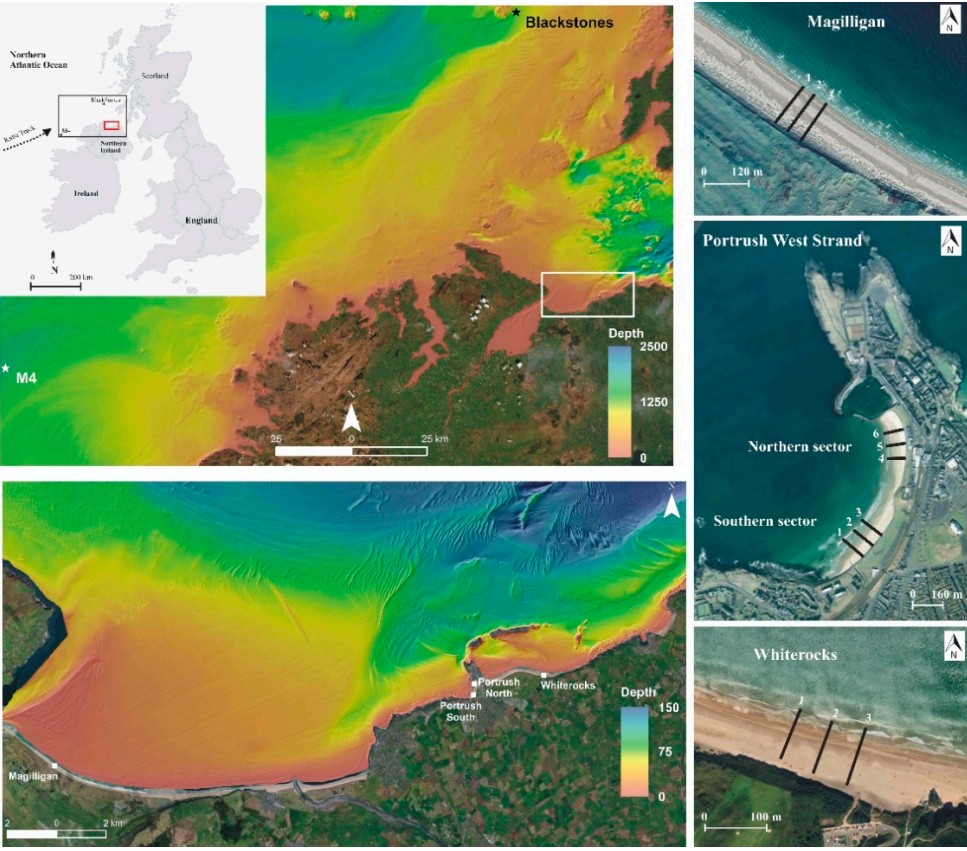

**Figure 2.** Location map with used grid for wave propagation and offshore buoys.

## 4. Methodology

A beach monitoring program was undertaken to investigate the impacts of storm Katia by surveying beach morphology before, during and after the storm. Surveys were conducted at Magilligan from 7 to 14 September 2011 while surveys at Portrush West Strand (at two sectors) and at Whiterocks beach took place between the 12 and 15 September 2011. Beach morphological changes were determined through topographic surveys extending from the back beach to low water level using a differential GPS (Trimble 4400) with 1–3 cm accuracy. Survey data were used to quantify the impact of Katia on beach morphology and volumetric changes at each sector. Cross-shore profiles were also extracted and their vertical morphological variability analysed to identify the main active zones [43,44].

Morphological and volumetric variations were compared with nearshore wave forcing to assess the process-response relationships in the monitored beaches, using the nearshore wave power, wave steepness and alongshore wave energy flux. These parameters have been extensively used for exploring morphological changes in wave-dominated beaches and found to be relevant indicators to understand beach erosion and accretion [45]. Here, we computed wave forcing indicators based on shallow water wave parameters obtained from high-resolution nearshore wave modelling using SWAN [46,47]. SWAN was implemented using a nested modelling scheme and forced in the western and northern boundaries with observed wave parameters measured at the M4 offshore buoy (55° N, 10° W), maintained by the Irish Marine Institute, and the Blackstones buoy (56°03′ N, 7°03′ W) operated by CEFAS (Figure 2). Waves were initially propagated over a large-scale computational grid with a resolution of 250 m and using the 2018 EMODnet bathymetry dataset [48] that extended from the buoy locations all the way into the north coast region (Figures 2 and 3a), in order to obtain the boundary conditions for a finer resolution run focused on the study area (Figures 2 and 3b). The model was run with an hourly timestep from 00:00 on the 7 September 2011 to 23:00 on the 16 September 2011. The nested nearshore wave runs were performed using a 5 m high-resolution computational

domain, implemented with a detailed bathymetric grid based on JIBS multibeam dataset for the North Coast (Figure 2). The nearshore runs were performed for the exact same time period indicated above, using wave spectra obtained from the large-scale run. The nested runs considered variable water levels obtained from the hourly records of Portrush Tide Gauge. SWAN was implemented in third generation, 2D stationary mode, using a JONSWAP spectral shape to represent the wave field, directional discretization in regular classes of 5° and frequency discretization in 33 logarithmic distributed classes between 1 and 0.03 Hz. Following Loureiro et al. [49] and Matias et al. [50], SWAN runs used default parameters for wave growth, whitecapping dissipation, depth-induced breaking according to the β-kd model for surf-breaking [51], triad and quadruplet wave-wave interactions. Outputs from SWAN provided wave conditions for the nearshore area in each survey site, extracted for a single point in front of the beach in 4 to 5 m water depth.

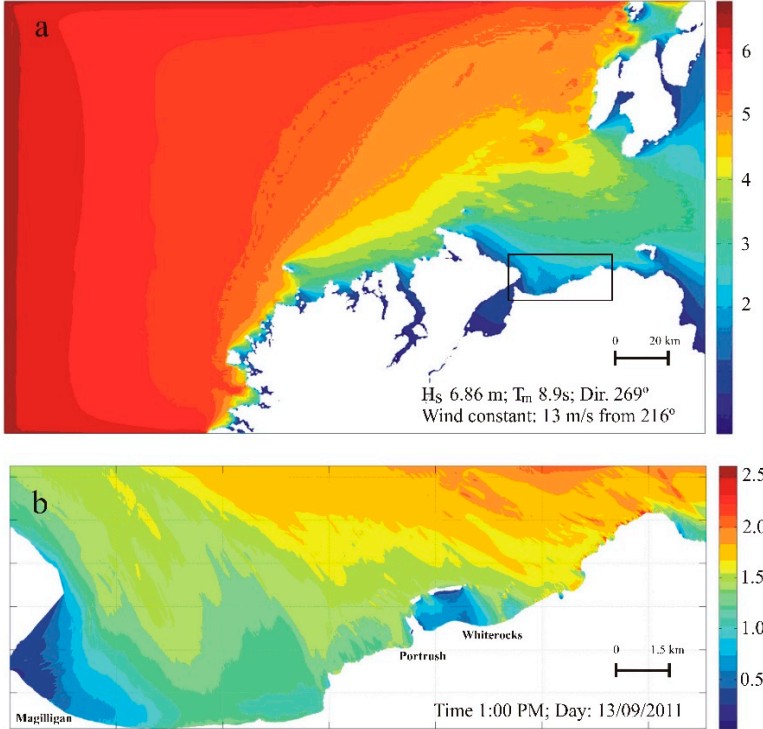

**Figure 3.** Modeled wave heights (m) for peak offshore storm conditions (**a**) and nearshore wave heights (m) (**b**) during above offshore wave conditions.

Based on SWAN outputs, wave forcing indicators for analyzing process-response relationships were computed assuming the shallow water approximations for linear wave theory following Komar [52]. Wave power ($P_s$) provides an indication of the rate at which energy is transferred by moving waves and is widely recognized as an important parameter for exploring wave-induced morphological change in sandy beaches (e.g., [45,52]). Wave power was computed according to:

$$P_s = EC_g \tag{1}$$

where E is wave energy, computed according to:

$$E = (1/8)\, \rho\, g\, H_s^2 \tag{2}$$

where $\rho$ is water density (1025 kg/m$^3$), g gravitational acceleration (9.81 m/s) and H$_s$ is nearshore wave height. Wave group velocity (C$_g$), was also obtained using the shallow water approximation:

$$C_g = \sqrt{g\,h} \tag{3}$$

where h is the water depth. Wave steepness (L$_s$) was determined according to:

$$L_s = T\,\sqrt{g\,h} \tag{4}$$

where T is the nearshore wave period.

Recognizing the importance of wave direction in combination with wave energy in driving longshore sediment transport and alongshore variable morphological changes during energetic conditions, particularly along indented or embayed coastlines (e.g., [45,53]), the alongshore component of the wave energy flux was also computed according to Komar's [52] approximation, given by:

$$P_l = P_s \sin \alpha_b \cos \alpha_b \tag{5}$$

where $\alpha_b$ is the wave breaking angle, determined according to the nearshore wave direction and beach orientation.

Wave steepness, obtained from the ratio of wave height (H$_s$) with wave length (L$_s$) was also computed for analyzing process-response relationships, considering the established association among high steepness waves (H/L > 0.02), offshore sediment transport and beach erosion in contrast to low steepness waves (H/L < 0.02), that are associated with onshore sediment transport and beach accretion [54].

## 5. Results

### 5.1. Wave Energy Spatial and Temporal Distribution

Wave conditions differed significantly between the offshore location where the wave records were obtained in the western coasts of Ireland and Scotland, and the nearshore areas adjacent to the monitored sites on the north coast of Northern Ireland (Figure 3).

Storm waves generated by the Katia post-tropical cyclone lasted until the 15 September, two days after the cyclone dissipated. Significant wave heights ranged from 1 to 3 m in the nearshore region, while offshore the wave buoys recorded values in excess of 5 m at the peak of the storm on 13 September 2011. Significant wave attenuation across the wide and irregular shelf and shoreface of Northern Ireland is evident from the exposed open ocean locations of the buoys to the relatively sheltered north coast area (Figures 2 and 3a,b). Considering in higher detail the variability within the north coast high resolution grid (Figure 3b), it is observed that wave heights also change significantly between the western, more protected area, towards the eastern more exposed one. Water levels recorded during the storm at Portrush's tidal gauge and modelled wave characteristics in each investigated site are presented in Figure 4. Maximum water levels were recorded on the 13 September 2011 (Figure 4a), with a storm surge effect ranging between 0.2 and 0.5 m induced by the low atmospheric pressure during the passage of the storm on the 13 September. Nearshore wave heights, even during the most energetic period of the storm, recorded between the 13 and 14 September and shown in Figures 3 and 4b, are relatively low, ranging from around 0.8 m in Magilligan to around 1.6 m in Portrush.

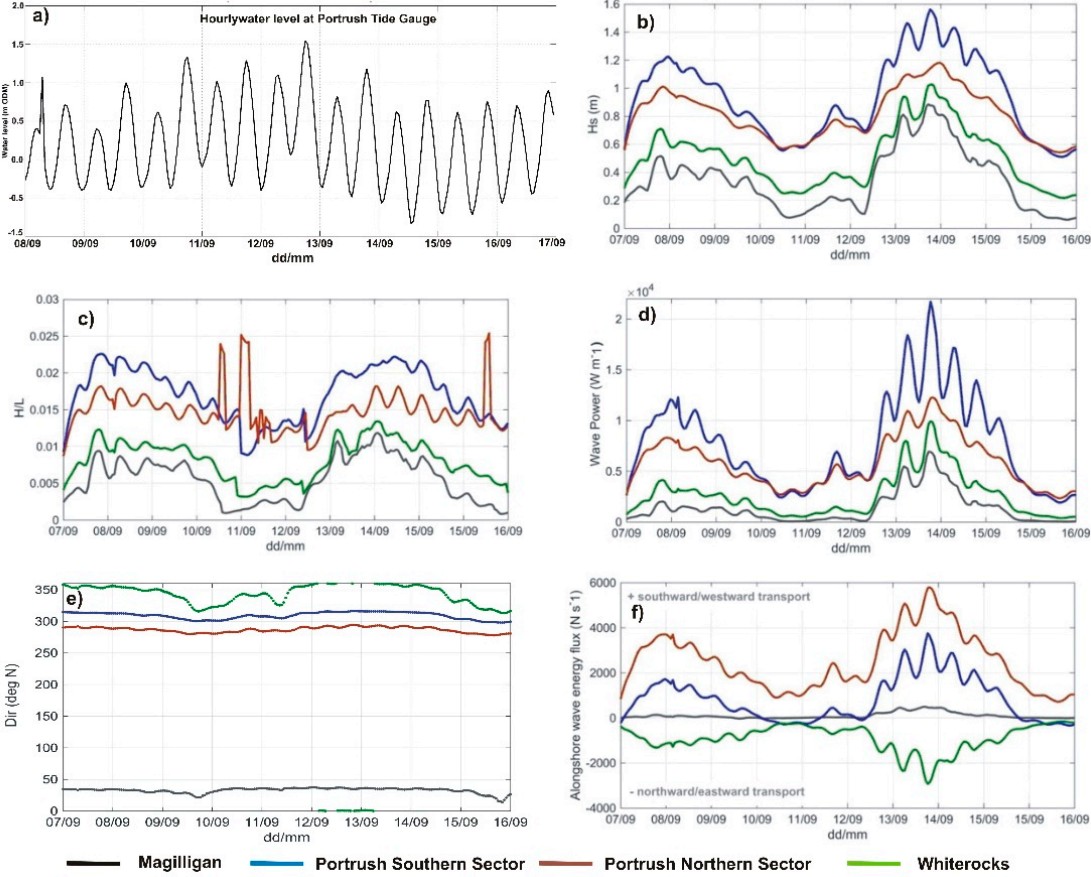

**Figure 4.** Water levels (**a**) at the Portrush tide gauge were reported as values above Ordnance Datum Malin (equivalent to Mean Sea Level). Wave height (**b**), wave steepness (**c**), wave power (**d**), wave direction (**e**) and alongshore wave energy flux (**f**) at different investigated sites.

This pattern is confirmed by the comparison of the time series for the different sites of interest; wave heights during storm Katia were not considerably high, having undergone significant attenuation and dissipation as they propagated through the shallow and irregular shelf of the north coast of Ireland. This is particularly noticeable for Magilligan, fronted by a wide and shallow shoreface, and Whiterocks, partially protected by the Skerries islands, while the more exposed Northern and Southern sectors of Portrush experienced more energetic conditions. Wave steepness (Figure 4c) displays significant spatial variability, while temporal changes largely reflected the variation in wave height during the surveyed period (Figure 4b). Lowest values were recorded in the most sheltered location, Magilligan, where the steepness ranged from ca. 0.002, during less energetic conditions (i.e., 11–12 September) to ca. 0.007 during more energetic conditions (i.e., 8–10 and 13–15 September periods). Steeper waves were observed at Portrush southern sector (Figure 4c), with values of ca. 0.015 and 0.025 for less and more energetic conditions, respectively. Such an increase in wave steepness, on the order of 0.05, from lower to moderate steepness conditions, was also observed in the other study sites (Figure 4c).

Spatial and temporal variability in wave power also reflected the changes in nearshore wave height, with distinct differences recorded among the different surveyed locations. The lowest values were observed at Magilligan whilst the highest were found at Portrush southern sector (Figure 4d). The dependence of nearshore wave power on water depth is particularly evident in Portrush southern sector, with clear temporal variation associated with the tide-induced changes in water level during the most energetic wave conditions. The influence of wave direction on morphological changes experienced in the four beaches during the storm, explored through the angle of approach and alongshore wave energy flux, indicate that relevant southward fluxes were experienced in the

more exposed Portrush Northern and Southern sectors, while at Whiterocks alongshore fluxes were easterly directed. In Magilligan, waves arrived fully refracted and shore normal, inducing negligible alongshore wave energy fluxes (Figure 4e,f).

### 5.2. Morphological and Volumetric Beach Changes

### 5.2.1. Magilligan

During the 7–10 September 2011, the beach showed limited elevation changes (on the order of 5–10 cm), which were largely uniform along both the cross-shore, i.e., along the dry beach and the foreshore and the longshore direction (Figure 5a). Volumetric variation indicates some general beach erosion (Table 2). The survey carried out on the 14th, i.e., after the most energetic waves impacted the beach (Figure 4b,d), showed longshore uniform erosion in the dry beach, with vertical erosion of ca. 10 cm, and an equivalent accretion at the central part of the beach according to a beach pivoting mechanism. Volumetric changes reflected a general, very small, accretion (Figure 5b, Table 2), with an alongshore uniform pattern consistent with the shore normal waves that impacted this beach during the storm (Figure 4e,f).

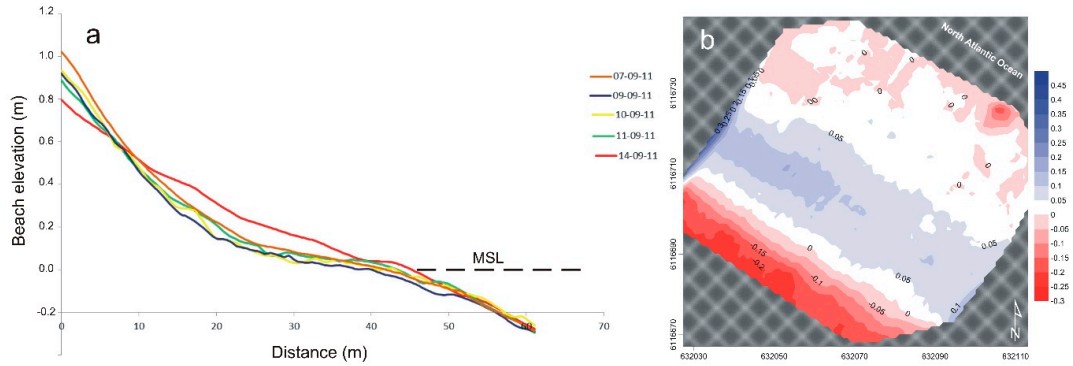

**Figure 5.** (**a**) Profile 2 evolution and (**b**) 3D morphological changes at Magilligan.

**Table 2.** Volumetric variation (m$^3$) between surveys and at the end of the monitoring program.

| Location/Date | 7 to 9 September | 9 to 10 | 10 to 11 | 11 to 14 | Whole Period 7th to 14th |
|---|---|---|---|---|---|
| Magilligan | −184.3 | +82.9 | +20.7 | +50.9 | +64.4 |
| **Location/date** | **12 to 13 September** | **13 to 14** | **14 to 15** | **-** | **Whole Period 12th to 15th** |
| Portrush southern | −1.1 | +512.3 | −226.9 | - | +123.9 |
| Portrush northern | −432.6 | +253.5 | +9.5 | - | −233.3 |
| Whiterocks | +316.5 | +158.6 | +109.2 | - | +623.2 |

### 5.2.2. Portrush, Southern Sector

During the initial phases of the storm, the beach presented very small morphological and volumetric changes (Figure 6 and Table 2) with a shore normal directed energy flux (Figure 4f). A uniform accretion along the cross-shore profile of ca. 10 cm was recorded on the 14 September (Figure 6a) and corresponded with a volumetric increase of 512.3 m$^3$ (Table 2). This was linked to the sediment supply from the Portrush northern sector due to northerly approaching waves that induced a southward directed flux (Figure 4f). During the last stages of the storm, approximately 5 cm of vertical erosion was observed in different parts of the profile, especially in the central and lower parts (Figure 6a and Table 2) probably due to the reduction of sediment inputs availability from the northern sector. Overall, from the 12 to 15 September, the beach presented a vertical accretion of 10–15 cm especially in the central-upper part (Figure 6b) associated with a volumetric accretion of 123.9 m$^3$, likely due to sediment supplied from the northern sector that recorded erosion (Figure 7).

The uppermost part of the beach showed different behavior, as its dynamic is strongly affected by a backing concave concrete seawall.

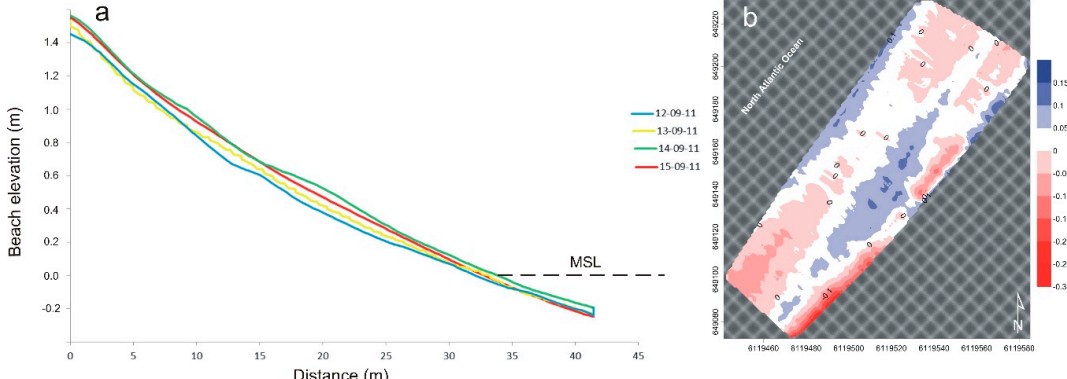

**Figure 6.** (**a**) Profile 2 evolution and (**b**) 3D morphological changes at Portrush Southern Sector.

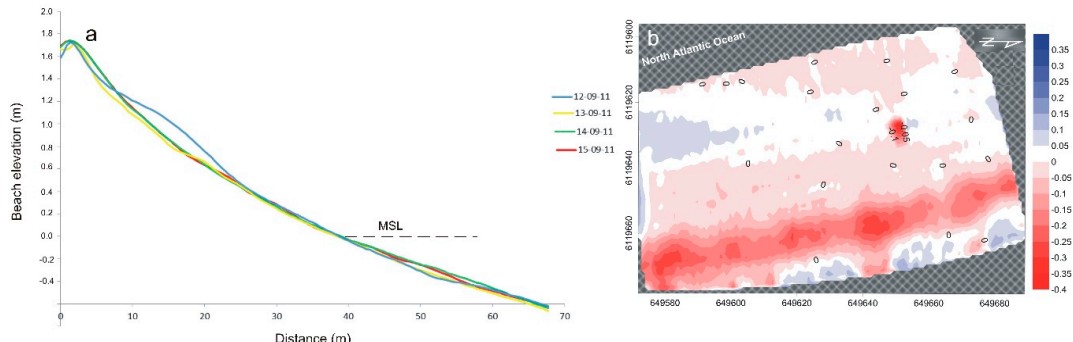

**Figure 7.** (**a**) Profile 2 evolution and (**b**) 3D morphological changes at Portrush Northern Sector.

5.2.3. Portrush, Northern Sector

The main morphological and volumetric changes were recorded in the first stages of the storm, i.e., on the 13th September (Figure 7a and Table 2). A well-developed berm, 20–30 cm in height present along the whole sector investigated, was completely eroded with a sediment loss of approximately 432.6 m$^3$ (Table 2). The berm was essentially transported southward to feed the southern sector of this pocket beach and, to a lesser extent, the middle and lower part of the profile (Figure 7a). This is the result of a relevant (2000 N/s) southward directed wave energy flux, forced by northerly approaching waves observed during that period (Figure 4e,f). In the following surveys only very small changes, of few centimetres, were observed. Overall, the beach recorded 30–35 cm of vertical erosion (Figure 7a,b) along the upper part, through the erosion of the berm, which corresponded to a volumetric change of −233.3 m$^3$ (Table 2).

5.2.4. Whiterocks

Prior to the storm, the beach presented a well-developed berm, ca. 30–40 cm in height (Figure 8a). During the first stages of the storm, on the 13 September, relevant morphological changes took place linked to the erosion of the berm and the landward and seaward transport of the eroded sand according to a process of beach flattening—but no net erosion was recorded (Table 2), this indicating a supply of sediments from the western part of Curran strand driven by eastward alongshore wave energy fluxes under low energy westerly approaching waves (Figure 4e,f). No relevant morphological changes took place in following days and accretion was recorded along all the profile (Figure 8a and Table 2). Overall, at the end of the storm, the beach recorded a volumetric accretion of 632.2 m$^3$ visible on the dry beach and at the lower foreshore (Figure 8b).

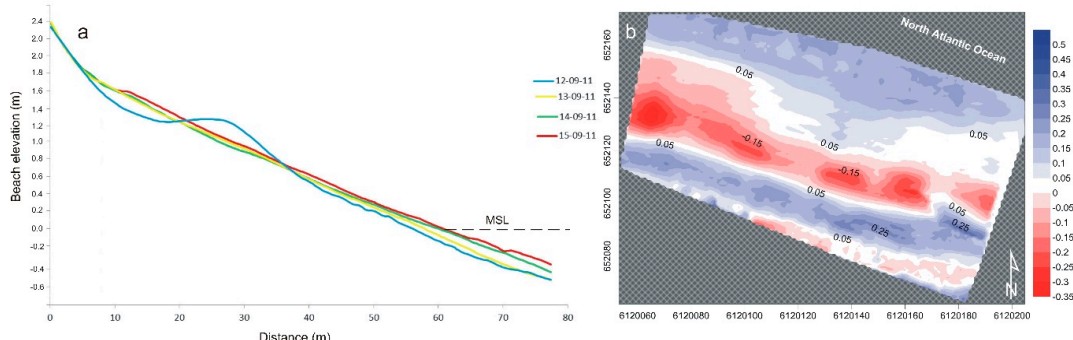

**Figure 8.** (**a**) Profile 2 evolution and (**b**) 3D morphological changes at Whiterocks.

## 6. Discussion

The Northern and Western coasts of Ireland are classified as high-energy coastlines and are often affected by energetic storms and the tail-end of a small number of Atlantic Hurricanes [32,33,55–57]. The effects of some of these events have been described by different authors [32,55–60], but the specific impacts of a recent post-tropical cyclone on Northern Irish beaches have not been investigated so far.

Coastal morphological changes investigated in this paper were related to the effects of the post-tropical cyclone Katia that impacted the coast of Northern Ireland from the 12 to the 15 September 2011 with sustained winds of ca. 95 km h$^{-1}$ and offshore waves in excess of 5 m in height. The cyclone originated as a tropical depression over the eastern Atlantic on 29 August, strengthened into a tropical storm the following day and than developed into a hurricane by 1st September, becoming a Category 4 hurricane with winds of 225 km h$^{-1}$ by 4 September moving eastward towards the east coast of USA and downgraded to a post-storm cyclone one day before reaching the British Isles on the 12–13 September 2011.

The earliest reported high-magnitude event in the Northwest of Ireland was the 'Night of the Big Wind', which was probably the tail-end of a hurricane that hit Ireland in January 1839. More recently, an analysis of instrumental storm records since the 1950s [60], identified Hurricane Debbie as the largest storm that impacted Ireland during the second half of the 20th Century [31,60,61]. Hurricane Debbie moved west across the Atlantic still maintaining hurricane-force winds and affecting Ireland in September 1961, with winds of up to 181 km h$^{-1}$ for more than 5 h along the west of Ireland [60]. Furthermore, Guisado-Pintado and Jackson [32,33] mentioned the effects of Hurricane Charley (1986), which was considered a post-tropical storm when it reached the south coast of Ireland, and described in detail the effects of the extratropical storm Ophelia. This event, re-classified as a "post-tropical" storm but being previously a Category 3 hurricane [61], reached the western coast of Ireland on the 16 October 2017, triggering a nationwide severe weather warning and causing substantial coastal flooding.

During storm Katia, offshore wave buoys recorded values in excess of 5 m at the peak of the event but nearshore significant wave height ranged from 1 to 3 m, reflecting relevant wave energy reduction linked to a variety of dissipation processes, that is: although offshore conditions in the exposed Atlantic section of the coast were very energetic, when waves propagate and refract to reach the study areas (Figure 2), they dissipate significant amounts of energy due to bottom friction, refraction and diffraction along the wide, shallow and irregular shelf and shoreface, and indented coastline [56]. This was particularly noticeable at Magilligan, as this area is protected towards the west by a resistant rocky headland and the shoreline is situated on the lee side of the Innishowen Peninsula that prevents the arrival of large amounts of wave energy from the SE, SW or NW quadrants [57]. Finally, at Magilligan, because of the wide shoreface adjacent to it and its long-term association with a large ebb-delta (Tuns Bank), bottom friction dissipation and refraction are more significant than at other sites examined here and the waves arrived almost perfectly shore parallel. This leads to a very small angle of approach and minimal alongshore wave energy fluxes. As a result, at Magilligan, there was limited wave forcing over the whole period, both in terms of wave height and power, but also reduced wave steepness

and almost insignificant alongshore fluxes. Specifically, with respect to other locations, wave power observed here was the lowest (i.e., $5 \times 10^3$ kW m$^{-1}$ during the peak of the storm), much smaller than what was observed during the last phases of Ophelia storm ($20 \times 10^3$ kW m$^{-1}$) [32]. As a result, only minor morphological changes were recorded at Magilligan with erosion of the upper beach and flattening of the foreshore, a trend similar—but with much smaller vertical morphological variations—to the one observed at Five Finger Strand in NW Ireland [32].

Less significant nearshore wave refraction favoured higher (compared to Magilligan) wave height and wave power values in Portrush and Whiterocks. Wave power ranged from 10 to $20 \times 10^3$ kW m$^{-1}$ and presented a clear longshore component, predominantly southward at both Portrush sectors and eastward at Whiterocks. Hence, longshore transport was evident at Portrush sectors that are included within a morphological sediment cell [9] enclosed by a harbour, in the northern end, and a headland in the southern end. The northern sector presented erosion while the southern sector accreted, probably from a point of pivoting in beach planform located in between the two sectors as observed at other morphological cells [9,62], which is consistent with a short-term rotational response identified in small embayments in various other settings [63–65] and in other studies of large embayed beaches [66–68].

At Whiterocks, a trend similar to the one noted at Portrush was observed. Berm erosion took place by means of a general beach flattening and this resulted in an overall accretion due to an influx of sediment from the western section of the beach.

Overall, storm Katia produced relatively limited impacts on the beaches of the north coast of Northern Ireland. This was also noted by Guisado-Pintado and Jackson [32] and Cooper et al. [56] that indicated that, to have a relevant impact on dissipative beaches of Northern Ireland, storms need to be directed onshore and coincide with high tide, rendering storm duration and offshore intensity of less importance. Katia, which reached approximately 96 km h$^{-1}$ presented much lower intensity that Debbie (181 km h$^{-1}$), but was marginally higher than Ophelia (gusts of 74 km h$^{-1}$) and had a longer duration than the previous events. This was not reflected in terms of morphological changes since during Katia the storm track and prevailing wind directions, which are relevant aspects in determining storm damages [23], were from the SW and W directions, while the beaches investigated are mainly exposed to N, NW and NE quadrants. Interestingly, this was also the case for previous hurricane/post-tropical cyclones Debbie and Ophelia. As observed by Cooper et al. [56], Northern Irish coasts facing NE and N are more susceptible to lower magnitude and longer duration storms, characterised by short sea waves, from a northerly direction. Further, as reported by Cooper et al. [56] and Guisado-Pintado and Jackson [32,33] and confirmed by this study, morphological changes produced on the coast were very localized and dependant on nearshore wave propagation driving cross and alongshore energy fluxes, since the volume and direction of transport during storm impact was highly site-specific. A similar trend, i.e., changes in the direction of longshore transport and morphological and volumetric (positive/negative) modalities of beach response to storms impacts, was recorded in SW England during the 2013–2014 storm winter season [69]. At Magilligan and Whiterocks, erosion processes did not greatly affect the dry beach and did not impact at all on the frontal dunes since this depends on storm peak coincidence with high tidal levels [32,56,65]. As observed by Cooper et al. [56] at Magilligan, the formation of the local dune escarpment is relatively rare and is typically associated with storms that occur at or close to high tide, with predicted tidal elevations of 2.1 m (high spring tide). During Katia storm, maximum water elevation was 1.4 m (high neap tide). The upper beach in Portrush, which is backed by a concrete concave seawall, experienced more relevant vertical changes, especially at the southern sector where the upper beach connects directly with the seawall and the backshore is inexistent.

## 7. Conclusions

Despite offshore wave buoys values in excess of 5 m wave height at the peak of the post-tropical cyclone Katia on the 13 September 2011, nearshore significant wave height ranged from 1 to 3 m. This was due to the dissipation processes experienced by waves as they propagate and refract along

a shallow and irregular shelf and shoreface to reach the studied coastal sectors. Such propagation process produced a very small angle of approach and minimal alongshore wave energy fluxes at Magilligan and, as a result, this location only recorded minor morphological changes with erosion of the upper beach and flattening of the foreshore. More limited nearshore wave refraction favoured higher waves at Portrush and Whiterocks, leading to increased wave power with a clear longshore component, ranging from southward at both Portrush sectors to eastward at Whiterocks. Longshore transport at Portrush favoured erosion in the northern sector and accretion in the southern sector, which is consistent with a short-term rotational response. At Whiterocks, a trend similar to the one noted at Portrush was observed, i.e., an overall accretion due to an influx of sediment from the western section of the beach.

Katia post-tropical cyclone, as other similar events—e.g., hurricane Debbie and post-tropical cyclone Ophelia, produced moderate impacts in the beaches investigated because the storm track and prevailing wind directions were from the SW and W directions, while the monitored beaches are mainly exposed to N, NW and NE quadrants. Overall, it is not possible to predict a general and homogeneous response of the north coast of Northern Ireland to such kind of events because morphological changes produced are very site-specific and dependant on water level during the storm and, especially, wave transformation across the shelf and shoreface that controls the volume and direction of sediment transport and hence, beach morphological response.

**Author Contributions:** Conceptualization, G.A., C.L. and D.J.; methodology, T.S. and C.L.; software, M.T.; investigation, G.A. and T.S.; resources, D.J.; data curation, C.L. and M.T.; writing—original draft preparation, G.A.; writing—review and editing, C.L., D.J. and T.S.; All authors have read and agreed to the published version of the manuscript.

**Funding:** This research received no external funding.

**Acknowledgments:** This work is a contribution to the Andalusia (Spain) PAI Research Group RNM-328. Carlos Loureiro contribution is developed in the framework of H2020 MSCA NEARControl project, which received funding from the European Union's Horizon 2020 Research and Innovation programme under the Marie Skłodowska-Curie grant agreement No. 661342.

**Conflicts of Interest:** The authors declare no conflict of interest.

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
