# Peer review of "Spatial Variability of Beach Impact from Post-Tropical Cyclone Katia (2011) on Northern Ireland’s North Coast"

_water, doi:10.3390/w12051380_

Round 1

Reviewer 1 Report

Dear author,

Thank you for the interesting article. There are some minor comments.

You well described wave properties during the Katrin, but nevertheless it is missing general information on SWL changes during the storm. And what was predominant wave direction for the yours analysed beaches. May you show wave energy spectral density for the selected points or at least predominant wave and wind directions and how it related to the coast (perpendicular to the coast or at some angle). That might explain why there were no strong erosion on the shore.

Figure 3 change in to the title to the wave energy flux.

I missed the conclusion section. And in general what are conclusions of the article?

Author Response

Observation:

Thank you for the interesting article. There are some minor comments.

You well described wave properties during the Katrin, but nevertheless it is missing general information on SWL changes during the storm. And what was predominant wave direction for the yours analysed beaches. May you show wave energy spectral density for the selected points or at least predominant wave and wind directions and how it related to the coast (perpendicular to the coast or at some angle). That might explain why there were no strong erosion on the shore.

Answer:

We clarifies all issues and added a figure regarding wave direction – figure 4 e.  

Figure 3 change in to the title to the wave energy flux.

We corrected it.

I missed the conclusion section. And in general what are conclusions of the article?

We added Conclusions.

Reviewer 2 Report

This study remains very insufficient from a scientific point of view; one has the impression of having a master's work not really deep. Therefore, this work, which is of definite interest, must therefore be improved.   1 - Cyclone Katia is absolutely not presented from a dynamic point of view (atmospheric and hydrodynamic). This element must be at least the subject of a full paragraph. This is all the more important as the authors compare this event to other historical events under discussion, but we have no elements of comparison. Moreover, the exact date of the cyclone, especially the day, is not given in the summary, in the introduction, or even in the methods. It is extremely annoying for the reader when it is necessary to readjust this episode in relation to the date of topomorphological measurements. 2 - Likewise, in FIG. 1, the path of the cyclone is presented in a too summary manner. A true cyclonic trajectory is made day by day, indicating the center of the depression (and the barometric pressure). These data exist in meteorological services that make these kinds of meteo-atmospheric measurements. 3 - The presentation of the sites also remains insufficient; i.e., an intermediate map between figure 1 (general map of the Angloxan Islands) and the 4 aerial photos of the beaches is missing. You need a regional map of Northern Ireland is required. Do these beaches really represent the coastal context of Northern Ireland as the title suggests? It also lacks a lot of morphological elements like bathymetry for example. This element is all the more important since the authors then discuss the effects of decreasing of wave energy due to the slope of the shoreface platform. In addition, bathymetry is also an important element of the SWAN modeling used by the authors. 4 - Methodological aspects, especially on SWAN modeling, must be improved. The authors present several equations, without justifying these choices. Why use the wave steepness (H/L)? Why use the alonshore wave energy flux when all the sites do not have longshore dynamics? The measurement campaigns from one site to another are not very consistent. For 3 sites, they start when the cyclone arrives on the study sites (September 12 to 13). 5 - The discussion remains insignificant. There has been a lot of work on the impact of storms along the Anglo-Saxon islands or Brittany. It is really necessary to replace this event within a multi-decennial chronology of extreme events, in particular in terms of intensity and morphological impacts. From the introduction, we must talk about winter 2013-2014 (Masselink et al., 2015). See and discuss also:
  - Olivier Burvingt, Gerd Masselink, Paul Russell, Tim Scott (2017) - Classification of beach response to extreme storms. Geomorphology 295 (2017) 722–737.
- Tim Poate, Gerhard Masselink, Paul Russell, Martin Austin (2014) - Morphodynamic variability of high-energy macrotidal beaches,
Cornwall, UK. Marine Geology 350 (2014) 97–111.   - Gerd Masselink, Bruno Castelle, Tim Scott, Guillaume Dodet, Serge Suanez, Derek Jackson, and France Floc’h (2016) - Extreme wave activity during 2013/2014 winter and morphological impacts along the Atlantic coast of Europe. Geophys. Res. Lett., 43, doi:10.1002/2015GL067492.   - Betts, N.L., Orford, J.D., White, D., Graham, C.J., 2004. Storminess and surges in the South-Western Approaches of the eastern North Atlantic: the synoptic climatology of recent extreme coastal storms. Mar. Geol. 210, 227–246. doi:10.1016/j.margeo.2004.05.010   - Blaise, E., Suanez, S., Stéphan, P., Fichaut, B., David, L., Cuq, V., Autret, R., Houron, J., Rouan, M., Floc’h, F., Ardhuin, F., Cancouët, R., Davidson, R., Costa, S., Delacourt, C., 2015. Bilan des tempêtes de l’hiver 2013-2014 sur la dynamique de recul du trait de côte en Bretagne. Géomorphologie Reli. Process. Environ. 21, 267–292. doi:10.4000/geomorphologie.11104   -Blender, R., Fraedrich, K., Lunkeit, F., 1997. Identification of cyclone-track regimes in the North Atlantic. Q. J. R. Meteorol. Soc. 123, 727–741. doi:10.1002/qj.49712353910   -Matthews, T., Murphy, C., Wilby, R.L., Harrigan, S., 2014. Stormiest winter on record for Ireland and UK. Nat. Clim. Chang. 4, 738–740.   -McCallum, E., Norris, W.J.T., 1990. The storms of January and February 1990. Meteorol. Mag. 119, 201–210.

-Gómara, I., Rodríguez-Fonseca, B., Zurita-Gotor, P., Pinto, J.G., 2014. On the relation between explosive cyclones affecting Europe and the North Atlantic Oscillation. Geophys. Res. Lett. 41, 2182–2190. doi:10.1002/2014GL059647

-Dissanayake, P., Brown, J., Karunarathna, H., 2015a. Impacts of storm chronology on the morphological changes of the Formby beach and dune system , UK. Nat. Hazards Earth Syst. Sci. 15, 1533–1543. doi:10.5194/nhess-15-1533-2015

-Dissanayake, P., Brown, J., Wisse, P., Karunarathna, H., 2015b. Effects of storm clustering on beach/dune evolution. Mar. Geol. 370, 63–75. doi:10.1016/j.margeo.2015.10.010

Author Response

This study remains very insufficient from a scientific point of view; one has the impression of having a master's work not really deep. Therefore, this work, which is of definite interest, must therefore be improved.   1 - Cyclone Katia is absolutely not presented from a dynamic point of view (atmospheric and hydrodynamic). This element must be at least the subject of a full paragraph. This is all the more important as the authors compare this event to other historical events under discussion, but we have no elements of comparison.

We added information and made a map.

Moreover, the exact date of the cyclone, especially the day, is not given in the summary, in the introduction, or even in the methods. It is extremely annoying for the reader when it is necessary to readjust this episode in relation to the date of topomorphological measurements. 

We added information.

2 - Likewise, in FIG. 1, the path of the cyclone is presented in a too summary manner. A true cyclonic trajectory is made day by day, indicating the center of the depression (and the barometric pressure). These data exist in meteorological services that make these kinds of meteo-atmospheric measurements. 

As above.

3 - The presentation of the sites also remains insufficient; i.e., an intermediate map between figure 1 (general map of the Angloxan Islands) and the 4 aerial photos of the beaches is missing. You need a regional map of Northern Ireland is required. Do these beaches really represent the coastal context of Northern Ireland as the title suggests? It also lacks a lot of morphological elements like bathymetry for example. This element is all the more important since the authors then discuss the effects of decreasing of wave energy due to the slope of the shoreface platform.

We added two bathymetric maps.

In addition, bathymetry is also an important element of the SWAN modeling used by the authors.4 - Methodological aspects, especially on SWAN modeling, must be improved. The authors present several equations, without justifying these choices. Why use the wave steepness (H/L)? Why use the alonshore wave energy flux when all the sites do not have longshore dynamics?The measurement campaigns from one site to another are not very consistent.

We clarified such issues.

 For 3 sites, they start when the cyclone arrives on the study sites (September 12 to 13). 

Yes Magilligan was surveyed for a longer time, the other three sites at the same time....It is well explained and I guess it is not a very relevant issue.

5 - The discussion remains insignificant. There has been a lot of work on the impact of storms along the Anglo-Saxon islands or Brittany. It is really necessary to replace this event within a multi-decennial chronology of extreme events, in particular in terms of intensity and morphological impacts. From the introduction, we must talk about winter 2013-2014 (Masselink et al., 2015). See and discuss also:

We added information.

  - Olivier Burvingt, Gerd Masselink, Paul Russell, Tim Scott (2017) - Classification of beach response to extreme storms. Geomorphology 295 (2017) 722–737. 
- Tim Poate, Gerhard Masselink, Paul Russell, Martin Austin (2014) - Morphodynamic variability of high-energy macrotidal beaches,Cornwall, UK. Marine Geology 350 (2014) 97–111.  

 - Gerd Masselink, Bruno Castelle, Tim Scott, Guillaume Dodet, Serge Suanez, Derek Jackson, and France Floc’h (2016) - Extreme wave activity during 2013/2014 winter and morphological impacts along the Atlantic coast of Europe. Geophys. Res. Lett., 43, doi:10.1002/2015GL067492.  

- Betts, N.L., Orford, J.D., White, D., Graham, C.J., 2004. Storminess and surges in the South-Western Approaches of the eastern North Atlantic: the synoptic climatology of recent extreme coastal storms. Mar. Geol. 210, 227–246. doi:10.1016/j.margeo.2004.05.010  

- Blaise, E., Suanez, S., Stéphan, P., Fichaut, B., David, L., Cuq, V., Autret, R., Houron, J., Rouan, M., Floc’h, F., Ardhuin, F., Cancouët, R., Davidson, R., Costa, S., Delacourt, C., 2015. Bilan des tempêtes de l’hiver 2013-2014 sur la dynamique de recul du trait de côte en Bretagne. Géomorphologie Reli. Process. Environ. 21, 267–292. doi:10.4000/geomorphologie.11104  

 -Blender, R., Fraedrich, K., Lunkeit, F., 1997. Identification of cyclone-track regimes in the North Atlantic. Q. J. R. Meteorol. Soc. 123, 727–741. doi:10.1002/qj.49712353910  

-Matthews, T., Murphy, C., Wilby, R.L., Harrigan, S., 2014. Stormiest winter on record for Ireland and UK. Nat. Clim. Chang. 4, 738–740.  

-McCallum, E., Norris, W.J.T., 1990. The storms of January and February 1990. Meteorol. Mag. 119, 201–210.

-Gómara, I., Rodríguez-Fonseca, B., Zurita-Gotor, P., Pinto, J.G., 2014. On the relation between explosive cyclones affecting Europe and the North Atlantic Oscillation. Geophys. Res. Lett. 41, 2182–2190. doi:10.1002/2014GL059647

-Dissanayake, P., Brown, J., Karunarathna, H., 2015a. Impacts of storm chronology on the morphological changes of the Formby beach and dune system , UK. Nat. Hazards Earth Syst. Sci. 15, 1533–1543. doi:10.5194/nhess-15-1533-2015

-Dissanayake, P., Brown, J., Wisse, P., Karunarathna, H., 2015b. Effects of storm clustering on beach/dune evolution. Mar. Geol. 370, 63–75. doi:10.1016/j.margeo.2015.10.010

Round 2

Reviewer 2 Report

The article has been significantly improved. The comments and requests for corrections that I made have been taken into account. The article may now be published.